# Scabies prevalence and its associated factors among prisoners in southern Ethiopia: An institution-based analytical cross-sectional study

Efa Ambaw Bogino[1], Beshada Zerfu Woldegeorgis[2], Lantesil Wondewosen[1], Blen Kassahun Dessu[3], Mohammed Suleiman Obsa[4], Lolemo Kelbiso Hanfore[5], Teketel Ermias Galtore[6], Woldu Kidane[7], Abraham Getachew Kelbore[1] *

1 Department of Dermatovenereology, Wolaita Sodo University, Wolaita Sodo, Ethiopia, 2 Department of Internal Medicine, Wolaita Sodo University, Wolaita Sodo, Ethiopia, 3 Department of Anesthesia, Wolaita Sodo University, Wolaita Sodo, Ethiopia, 4 Department of Anesthesia, Arsi University, Assela, Ethiopia, 5 School of Nursing, Wachemo University, Hossana, Ethiopia, 6 Department of Midwifery, College of Health Sciences and Medicine, Wachemo University, Durame, Ethiopia, 7 AMREF Health Africa Ethiopia, Addis Ababa, Ethiopia

* kelbore2005@gmail.com

**Data Availability Statement:** All relevant data are within the manuscript. Additional data inquiries may be directed to Chief of Research &

## Abstract

### Background

Scabies is an ectoparasitic infestation that can easily spread during close human contact and in overcrowded areas such as prisons and public places where sanitation is a problem. Globally, as many as 300 million people are infested with scabies each year. Within Ethiopia, its burden in institutions like prisons is not well-studied. As a consequence, we sought to estimate the prevalence of scabies and also identify factors associated with it among prison inmates in southern Ethiopia.

### Methods

An institution-based analytical cross-sectional study design was employed. We used, a simple random sampling technique to select 422 prisoners. A pretested-structured questionnaire was used to collect the necessary information. EpiData version 4.2.0.0 and Statistical Packages for Social Science version 25 software were used for data entry and analysis respectively. Both descriptive and inferential statistics were computed. The model fitness was checked using the Hosmer-Lemeshow and omnibus tests. The association between the independent and outcome variables was presented in the form of a table showing the crude odds ratio and adjusted odds ratio (AOR) along with their 95% confidence interval (CI). The level of statistical significance was declared at $P$.value $\leq$ 0.05.

### Results

A total of 418 prisoners were interviewed, yielding a 99.1% response rate. The age of the participants ranged from 17 to 60 years. As many as 381 (91.1%) participants were male.

Community Service director office
(chsm_rcttd@wsu.edu.et), College of Health
Sciences and Medicine, Wolaita Sodo University.

**Funding:** The author(s) received no specific
funding for this work.

**Competing interests:** The authors have declared
that no competing interests exist.

The prevalence of scabies was found to be 8.9%. A history of sexual contact in the past two months, before imprisonment (AOR: 9.92 (95% CI 3.07, 32.02), $P$ <0.001), a period of stay less than two months duration in the prison (AOR: 4.54 (95% CI 1.51, 13.54), $P$ = 0.007), poor ventilation (AOR: 3.36 (95% CI 1.07, 10.58), $P$ = 0.038), lack of hygiene soap (AOR: 5.53 (95% CI 1.45, 21.17), $P$ = 0.012), and sharing clothes among each other (AOR: 3.81 (95% CI (1.09, 13.29), $P$ = 0.036) were factors independently associated with a scabies infestation.

## Conclusion

In this study, we found the prevalence of scabies was high among prison inmates in Wolaita Zone prison. Furthermore, scabies infestations were associated with a previous history of sexual contact, poorly ventilated rooms, lack of hygiene soaps for washing, and sharing clothes amongst the inmates. Prison administrations should be encouraged to improve sanitary and screening and isolation of prisoners during imprisonment.

### Author summary

This article analyses the burden of Scabies and its associated factor among prison inmates at Wolaita Zone, Southern Ethiopia. There has been limited evidence how widespread Scabies is in Prison inmates in Ethiopia. Local Tropical dermatologists carried out skin examinations of Prison inmates in one prison clinic at Wolaita zone. All sampled prisoners were screened and diagnosed according to the International Alliance for Control of Scabies (IACS) criteria. During the study period those who diagnosed with scabies were treated accordingly. Our study showed the prevalence of scabies was high among prison inmates. Furthermore, the current study identified clothe sharing, not use of soap, and a history of sexual contact were factors independently associated with scabies infestation among prison inmates. In conclusion, proper intervention measures could help to prevent and control scabies in prison, but also to reduce the widespread of the diseases, transmission to the community and complication of this treatable condition.

## Introduction

Scabies is a highly contagious skin condition caused by Sarcoptes scabiei var hominis mites, which causes health problems in resource-scarce countries [1]. It causes intense itching and scratching that often leads to serious skin infections [2]. It is usually spread by direct skin-to-skin contact with a person who has scabies [3]. The disease can affect anyone from all socio-economic levels, with no preference to age, sex, race, or standards of personal hygiene. And spreads rapidly in closed communities through crowded conditions like in prisons [4].

   Globally, scabies is a common public health problem that affects more than 300 million people per year [5]. A systematic review of population-based studies from various regions of the world(excluding North America) found the highest prevalence of scabies in the Pacific region and Latin America ranging from 0.2–71% [6]. A study in Brazil showed the prevalence of scabies to be 9.8% [7]. In Malaysia, it was 8.1% and in Indian prisoners, 1.8% [8], and a systematic review in Ethiopia showed a prevalence of 14.5% [9].

In low and moderate- income countries, the sheer burden of scabies infestations and its complications imposes a major burden on healthcare systems and causes 0.21% of total disability-adjusted life years lost [10]. In sub-Saharan Africa, the scabies burden varies among countries. In Nigeria, the prevalence of the disease was found in a study to be 65% [11]. Evidence from the western region of Cameroon suggests that the prevalence of scabies was 32.1% among prisoners [12]. Similarly, community-based studies in Ethiopia showed the prevalence of scabies varies with the highest prevalence found in northwest Ethiopia's Amhara region78.4% [13]. The study mentioned before from Nigeria showed that clothes/bed-sharing, duration of confinement and low educational level were significantly associated factors [11]. the study of the Cameroon prisoners also identify overcrowding and sharing of clothes or beds as significant factors while regular bathing and soap use seemed to be protective against scabies infestation [12].

Community-based studies conducted in Ethiopia had similar results: sharing clothes with others, family history of scabies, poor hand washing practice, infrequent use of soap or not using soap for showering, not cleaning the room/house every day and the presence of pet animals in the home were factors associated with scabies infestation [14–18].

Although various Ethiopian community-based studies have been conducted to show the burden of scabies, no study shows the disease burden at prisons. In low setting like our, the prison is one of the overcrowded areas that are prone to the spread of contagious diseases like scabies, which can lead to complications. Thus, this study aimed to assess scabies prevalence and associated factors among prison inmates in the Wolaita zone, Ethiopia, to be used as baseline information for other researchers in and outside the country.

## Methods

### Ethics statement

Ethical clearance was obtained from the ethical review committee of the College of Health Sciences and Medicine, Wolaita Sodo University with reference number CARD 865/869/12. After permission was obtained from the prison administrator to conduct the research, the participants provided written informed consent to conduct the interview and physical examination. During the data collection, study participants whom the clinical diagnosis of scabies ascertained were treated with 5% permethrin cream.

### Study design

An institution-based analytical cross-sectional study design was employed.

### Setting

The study was conducted in the Wolaita zone prison, which is found in the Wolaita Sodo town, 329 kilometers from Ethiopia's capital, Addis Ababa. During the data collection, there were 1600 prisoners in the institution. The Wolaita Zone health department regularly monitors and supports the prison's clinic, which has been offering outpatient and emergency services for prison inmates around-the-clock since it was established in 1929 Ethiopian calendar. The study was conducted from 1st September to 30th December 2020.

### Participants

All prisoners in the institution were considered as source population. Individuals who were severely ill and had depression during the data collection were excluded from the study while those randomly selected were interviewed.

## Sample size determination

The required sample size for the study was directly computed by using Open Epi version 3.01 based on the following assumptions: 5% alpha, 80% power, and the proportion of 0.50 because no related previous study existed. After adding the non-response rate of 10%, the final sample size became 422.

## Sampling procedure

The study subjects were selected by using a simple random sampling method. For this purpose, the lists of all prisoners' numbers were used as the sampling frame. The total numbers of prison inmates were 1600. Then 422 prison inmates were randomly selected by Computer Microsoft Excel Random sampling from the lists. At the institution's prison clinic, all randomly chosen participants who volunteered for the study underwent a clinical examination.

## Case definition

The diagnosis of scabies was based on clinical evaluation according to the International Alliance for Control of Scabies (IACS) criteria. So at least the presence of one of the following was used for the diagnosis of scabies. Scabies burrows, typical lesions in a typical distribution (a scratch mark or vesicles or excoriated popular lesions on the wrist, infra-mammary, interdigital web spaces, and male genital area) [19].

## Measurement and data quality assurance

A structured questionnaire was used to collect all relevant data. Furthermore, the morphology of lesion characteristics was obtained during the clinical examination of the inmate at the prison clinic. The interview was conducted after their clinical examination. Four health professionals were involved in the data collection process. Two tropical dermatology specialists conducted the clinical examination. They used the IACS diagnostic criteria for scabies diagnosis and ruled out secondary infections. Two Bachelor science degree holding nurses were involved as interviewers. The interviews were carried out using a structured questionnaire. A pre-test was performed at Wolaita Sodo town jail on 5% of the total sample (n = 21) and then some modifications to the tool were made. Data completeness and consistency were checked using a simple frequency during the data entry and cleaning process.

## Statistical method

Data were entered into EpiData software version 4.2.0.0 and exported to Statistical Packages for Social Science version 25 for analysis. A descriptive analysis using proportion, mean, standard deviation, and range was computed. Initially, bivariable logistic regression analyses were run, and variables found to be significant at *P* value <0.25 were fitted to the multivariable binary logistic regression. The Hosmer-Lemeshow statistic method was checked for model fitness, and it was fit. A multiple logistic regression model was used to identify the association of independent variables with the outcome variable, and a *p*-value < 0.05 was used as a cut of point to declare the statistical significance. Adjusted odds ratio (AOR) along with their 95% confidence interval (CI) was also used to how the strength of association.

**Table 1. Socio-demographic characteristics of prison inmates in Wolaita zone prison, southern Ethiopia, 2020 (n = 418).**

| Variables | Category | Frequency | Percent |
|---|---|---|---|
| **Gender** | Male | 381 | 91.1 |
| | Female | 37 | 8.8 |
| **Residence before imprisonment** | Rural | 350 | 83.7 |
| | Urban | 68 | 16.3 |
| **Educational Level** | No formal Education | 75 | 17.9 |
| | Primary | 193 | 46.2 |
| | Secondary | 124 | 29.7 |
| | Above secondary | 26 | 6.2 |
| **Occupation** | Student | 100 | 23.9 |
| | Driver | 41 | 9.8 |
| | Farmer | 145 | 34.7 |
| | Merchant | 54 | 12.9 |
| | Civil servant | 26 | 6.2 |
| | Others | 52 | 12.4 |

## Results

### Socio-demographic characteristics of the participants

A total of 418 prisoners were examined with a response rate of 99.1%. Of these, 381(91.1%) were male. The age of the participants ranged from 17 to 60 years. Three hundred fifty (83.7%) of the study participants were living in rural areas before imprisonment, and 343 (82.1%) were able to read and write (Table 1).

### Environmental, personal hygiene, and clinical-related factors

The average length of stay in prison was 17.9 ± 18.8 months. The number of individuals in one room ranged from 5 to 326 prisoners. Furthermore, 352 (84.2%) reported that their room was cleaned daily and 384 (91.9%) reported that their room was well-ventilated. According to reports from 412 (98.6%) respondents, water was available daily in the compound, and 242 (57.9%) of them took a body shower every other day and 416 (99.5%) of study subjects used soap.

### Prevalence of scabies among prison inmates

In this study, we found that approximately one in eleven (8.9%) prisoners had scabies infestation. An intensively itchy sensation of the skin which was aggravated during the night was experienced by almost all the study participants who were diagnosed with scabies. Regarding the clinical presentation, the most common skin manifestation was an excoriated papular lesion, mostly seen on the fingers, axillar and genital area (97.3%). When asked about related problems in their rooms, 15 (40.5%) of them report a history of similar itching in their room, while 22 (59.5%) did not. Moreover, a history of sexual contact in recent two months before the appearance of the lesion was reported in 7(36.8%) of prisoners who had scabies, and 7 (35%) reported that they had shared personal clothes with others in the room.

### Factors associated with scabies among prison inmates

A bivariable binary logistic regression analysis was carried out to assess the association between scabies and the independent variables. Among the independent variables, sharing

**Table 2. Factors associated with scabies among prisoner inmates in Wolaita Zone prison, southern Ethiopia, 2020 (n = 418).**

| Variable | Category | Scabies | | Bivariate analysis | Multivariate analysis | |
|---|---|---|---|---|---|---|
| | | Yes, n(%) | No, n(%) | COR (95%CI) | AOR (95%CI) | P-Value |
| **Clothes sharing** | Yes | 7(35) | 13(65) | 6.61(2.45, 17.80) | 3.81(1.09, 13.29) | 0.036* |
| | No | 30(7.5) | 368(92.5) | 1 | 1 | |
| **Daily room cleaning** | Yes | 27(7.7%) | 325(92.3) | 1 | 1 | 1 |
| | No | 10(15.2) | 56(84.8) | 2.15(0.99, 4.68) | 1.90(0.746, 4.835) | 0.179 |
| **Duration in prison** | ≤2 months | 7(18.4) | 31(81.6) | 2.63(1.07, 6.49) | 4.527(1.51, 13.54) | 0.007* |
| | >2 months | 30(7.9) | 350(92.1) | 1 | 1 | 1 |
| **History of sexual contact** | Yes | 7(36.8) | 12(63.2) | 7.18(2.63, 19.58) | 9.92(3.07, 32.02) | <0.001* |
| | No | 30(7.5) | 369(92.5) | 1 | 1 | 1 |
| **Soap use** | Yes | 31(07.7) | 372(92.3) | 1 | 1 | 1 |
| | No | 6(40) | 9(60) | 8.00(2.67, 23.94) | 5.53(1.45, 21.17) | 0.012* |
| **Daily water availability** | Yes | 30(7.8) | 353(92.2) | 1 | 1 | 1 |
| | No | 7(20) | 28(80) | 2.94(1.19, 7.30) | 1.73(0.47, 6.42) | 0.41 |
| **Educational status** | No formal Education | 7(8.6) | 74(91.4) | 2.11(0.62, 7.26) | 1.04(0.24, 4.54) | 0.952 |
| | Primary | 19(9.9) | 172(90) | 1.81(0.62, 5.28) | 0.94(0.25, 3.48) | 0.920 |
| | Secondary | 6(5.2) | 110(94.8) | 3.67(1.04, 12.98) | 1.93(0.43, 8.61) | 0.390 |
| | More than Secondary | 5(16.7) | 25(83.3) | 1 | 1 | 1 |
| **Frequency of cloth change** | Daily | 7(24.1) | 22(75.9) | 1 | 1 | 1 |
| | Every two days | 23(8.7) | 242(91.3) | 3.35(1.30, 8.67) | 2.34(0.67, 8.14) | 0.183 |
| | Every three days | 7(5.6) | 117(94.4) | 5.32(1.70, 16.67) | 3.81(0.91, 15.98) | 0.068 |

1, Reference category;

*, Statistical significance at P<0.05

clothes, lack of daily room cleaning, less than two months duration of stay in prison, history of recent sexual contact within two months, poor room ventilation, lack of frequent soap use, unavailability of water daily, educational status, frequency of clothing change and the number of occupants in a single room were candidate variables for a multivariable logistic regression model. In the multivariable binary logistic regression analysis, a history of sexual contact in the previous two months, being a recent prisoner (less than two months in prison), poor ventilation in the room, lack of soap for showering, and sharing clothes with mates were variables that had a statistically significant association with a scabies.

Prisoners who had a history of recent sexual contact had 10 times [AOR: 9.92 (95% CI 3.07, 32.02), P<0.001] higher odds of developing scabies as compared to those who didn't. In addition, prisoners who had recently entered the prison (less than two months duration) had 5 times [AOR: 4.53 (95% CI 1.51, 13.54), P = 0.007] higher odds of developing scabies compared to those who stayed longer than two months in the prison. Prisoners who did not use soap had 6 times [AOR: 5.53(95% CI 1.45, 21.17), P = 0.012] higher odds of developing scabies when compared to those who frequently used soap. Lastly, prisoners who shared clothes with inmates had 4 times [AOR: 3.81(95% CI 1.09, 13.29), P = 0.036] higher odds of developing scabies compared to those who did not (Table 2).

## Discussion

This study aimed to determine the prevalence of scabies and identify factors associated with it among prison inmates. We found that approximately one in eleven (8.9%) prisoners had scabies infestation. Our finding was consistent with studies conducted in Malaysia (8.1%) [2,8],

and Brazil (9.8%) [4]. However, it was lower than studies from northern Ethiopia, Gonder (78%) [13], and Gojjam (35%) [18]. This difference could be because the later studies were conducted during the epidemic investigation. Furthermore, the observed difference might be due to a variation in population. The proportion of prisoners with scabies in this study was much higher than in studies conducted in Ghazel Hazar prison in Iran (2.2%) [20] and in the "Ankober" district of central Ethiopia (1.7%) [21]. The reasons might be due to differences in socio-economic conditions, sample size and duration of the study. Furthermore, the Ankober study was done in a large community during a mobile clinic.

This study also identified cloth sharing, soap use, history of sexual contact and duration of stay in prison as associated factors of scabies. Individuals who had shared their clothes had four times higher odds of developing scabies. The current finding was consistent with previous studies conducted in Ethiopia: Hadiya Zone Badewacho district [22], Wadila district [14]; Cameron [12], and Pakistan [23].

In addition, prisoners who did not use soap during handwashing had 6 times higher odds of exhibiting scabies. This finding was in line with other studies conducted in Ethiopia, Cameron, and Pakistan [12,14,23]. This could be due to improper handwashing practice or poor personal hygiene correlated with scabies infestation. Lastly, this study also identified a history of sexual contact in the previous two months to be significantly associated with sharing scabies. This finding is consistent with a review by Engelman et al [24]. This could be explained by scientific backgrounds that suggest intimate/direct body contact during sexual contact increases the risk of scabies infestation. We also acknowledged the limitations of this study. The study was conducted in one prison and most participants were males which limits generalization. Conditions related to sanitation in the prison could be not necessarily the same in all prisons.

## Conclusion and recommendations

In this study, the prevalence of scabies was high among prison inmates and the occurrence of scabies was associated with clothe sharing, not use of soap, and a history of sexual contact. Proper intervention measures could help to prevent and control scabies in prison. we recommend public health interventions must be taken; prison administrations should be encouraged to improve sanitary and screening and isolation of prisoners during enrollment and discharge helps to halt the spread of scabies in the community.

## Acknowledgments

We would like to extend our gratitude to Dr.Margaret G Matthews for editing the English grammar and we thank study participants, prison adminstrators and Wolaita sodo University for allowing us to conduct the study.

## Author Contributions

**Conceptualization:** Efa Ambaw Bogino, Abraham Getachew Kelbore.

**Data curation:** Blen Kassahun Dessu, Mohammed Suleiman Obsa, Abraham Getachew Kelbore.

**Formal analysis:** Efa Ambaw Bogino, Beshada Zerfu Woldegeorgis, Mohammed Suleiman Obsa, Lolemo Kelbiso Hanfore, Woldu Kidane, Abraham Getachew Kelbore.

**Investigation:** Efa Ambaw Bogino, Lantesil Wondewosen, Woldu Kidane.

**Methodology:** Efa Ambaw Bogino, Beshada Zerfu Woldegeorgis, Blen Kassahun Dessu, Woldu Kidane.

**Project administration:** Blen Kassahun Dessu.

**Supervision:** Efa Ambaw Bogino, Lantesil Wondewosen, Teketel Ermias Galtore.

**Validation:** Efa Ambaw Bogino, Woldu Kidane.

**Writing – original draft:** Efa Ambaw Bogino, Lantesil Wondewosen, Lolemo Kelbiso Hanfore, Abraham Getachew Kelbore.

**Writing – review & editing:** Efa Ambaw Bogino, Beshada Zerfu Woldegeorgis, Lantesil Wondewosen, Blen Kassahun Dessu, Mohammed Suleiman Obsa, Lolemo Kelbiso Hanfore, Teketel Ermias Galtore, Woldu Kidane, Abraham Getachew Kelbore.

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
