## [Decision Letter · Decision Letter 0]

9 Jun 2023

Dear Mr. Kelbore,

Thank you very much for submitting your manuscript "Scabies prevalence and associated factors among prisoners in Southern Ethiopia" for consideration at PLOS Neglected Tropical Diseases. As with all papers reviewed by the journal, your manuscript was reviewed by members of the editorial board and by several independent reviewers. The reviewers appreciated the attention to an important topic. Based on the reviews, we are likely to accept this manuscript for publication, providing that you modify the manuscript according to the review recommendations. 

Please address the comments raised by the reviewers.

Sincerely,

Katja Fischer

Academic Editor

Nigel Beebe

Section Editor

Please address the comments raised by the reviewers.

Reviewer's Responses to Questions

**Key Review Criteria Required for Acceptance?**

**Methods**

-Are the objectives of the study clearly articulated with a clear testable hypothesis stated?

-Is the study design appropriate to address the stated objectives?

-Is the population clearly described and appropriate for the hypothesis being tested?

-Is the sample size sufficient to ensure adequate power to address the hypothesis being tested?

-Were correct statistical analysis used to support conclusions?

-Are there concerns about ethical or regulatory requirements being met?

Reviewer #1: I can not judge the statistics. The objective and methods are clear.

Reviewer #2: (No Response)

**Results**

-Does the analysis presented match the analysis plan?

-Are the results clearly and completely presented?

-Are the figures (Tables, Images) of sufficient quality for clarity?

Reviewer #1: The results are clearly presented.

Reviewer #2: (No Response)

**Conclusions**

-Are the conclusions supported by the data presented?

-Are the limitations of analysis clearly described?

-Do the authors discuss how these data can be helpful to advance our understanding of the topic under study?

-Is public health relevance addressed?

Reviewer #1: Conclusions are supported by the data.

Reviewer #2: (No Response)

**Editorial and Data Presentation Modifications?**

Reviewer #1: see below

Reviewer #2: (No Response)

**Summary and General Comments**

Reviewer #1: Dear Authors,

I read your article with interest. Scabies is a problem in so many societies. Though a lot is known to control it, real control seems not possible as yet. I hoped you would contribute new information. But most were confirmative. However, a prison is a good place to do a study since it has well-defined circumstances. I hope that when you make some internal correlations, you may find certain directions where you may do further research in order to really contribute. Now your data are of regional importance.

Some remarks:

Introduction;

Probably better: Start a new sentence. The mites cause health problems ( instead of Pest problems ) in resource-scarce countries.

I would say that often leads to serious skin infections (Without “often” it is too absolute)

It will be nice to mention for each study whether it is done in communities or in prisons 

Methods:

In measurement I suppose you looked for the typical leasions in order to diagnose.

Case definition could be better placed before measurement. Then you do not have to repeat in this section.

• The first mention of the genital area should be deleted. 

Results:

4 refused. You did not replace them to get your 422? 

Table 1

In the frequency of gender is written a percentage of what? 

There was one room with 326 inmates. Did you look separate to this room for interest? I suppose this room was cleaned daily. Did this fit with the responses? 

Table 3

Daily room cleaning claimed by 325. How many rooms are there in total?

Is there a relationship between sex in the last two months and the duration of the stay in prison? 

Is there one factor related to higher educated having more scabies? ( More frequently sex?) Maybe just in this prison.

Interesting the indication that more frequent cloth changes lead to more scabies. How are the bedclothes cleaned? Are they shared? 

I wondered what is meant by room cleaning. Scabies does not survive on the floor. Do they sleep always in the same bed? And do bedclothes changing belong to daily room cleaning?

Reviewer #2: Study of scabies prevalence and risk factors in a prison in southern Ethiopia. The results are not new, but in this study it is related to a specific patient population.

The results should be published, however I still have some questions/comments:

1. there are some formal errors that should be corrected.

2. page 6, it is usually spread.....other ways of transmission should be mentioned

3. abbreviations LIC's, MIC's need to be explained.

4. page 7, were patients only informed verbally, no written informed consent?

5. page 8, were the mites or feces or eggs not detected by microscopical or dermatoscopical investigation to make the exact diagnosis?

PLOS authors have the option to publish the peer review history of their article (what does this mean?). If published, this will include your full peer review and any attached files.

Reviewer #1: No

Reviewer #2: No

Figure Files:

Data Requirements:

Reproducibility:

References

---

## [Editor Report · Decision Letter 1]

28 Nov 2023

Dear Mr. Kelbore,

We are pleased to inform you that your manuscript 'Scabies prevalence and its associated factors among prisoners in southern Ethiopia: An institution-based analytical cross-sectional study' has been provisionally accepted for publication in PLOS Neglected Tropical Diseases.

Best regards,

Nigel Beebe, PhD

Section Editor

Nigel Beebe

Section Editor

---

## [Editor Report · Acceptance letter]

14 Dec 2023

Dear Mr. Kelbore,

We are delighted to inform you that your manuscript, "Scabies prevalence and its associated factors among prisoners in southern Ethiopia: An institution-based analytical cross-sectional study," has been formally accepted for publication in PLOS Neglected Tropical Diseases.

Best regards,

Shaden Kamhawi

co-Editor-in-Chief

Paul Brindley

co-Editor-in-Chief
